# Chemistry and Biochemistry Aspects of the 4-Hydroxy-2,3-trans-nonenal

**DOI:** 10.3390/biom12010145

**Published:** 2022-01-16

**Authors:** Anna Bilska-Wilkosz, Małgorzata Iciek, Magdalena Górny

**Affiliations:** Chair of Medical Biochemistry, Faculty of Medicine, Medical College, Jagiellonian University, 7 Kopernika Street, 31-034 Krakow, Poland; miciek@cm-uj.krakow.pl (M.I.); mbgorny@cyf-kr.edu.pl (M.G.)

**Keywords:** 4-hydroxy-2,3-trans-nonenal, lipid peroxidation, Michael adducts, Schiff bases, hemiacetals, cancer, mental and psychosomatic disorders, anticancer activity

## Abstract

4-hydroxy-2,3-trans-nonenal (C_9_H_16_O_2_), also known as 4-hydroxy-*2E*-nonenal (C_9_H_16_O_2_; HNE) is an α,β-unsaturated hydroxyalkenal. HNE is a major aldehyde, formed in the peroxidation process of ω-6 polyunsaturated fatty acids (ω-6 PUFAs), such as linoleic and arachidonic acid. HNE is not only harmful but also beneficial. In the 1980s, the HNE was regarded as a “toxic product of lipid peroxidation” and the “second toxic messenger of free radicals”. However, already at the beginning of the 21st century, HNE was perceived as a reliable marker of oxidative stress, growth modulating factor and signaling molecule. Many literature data also indicate that an elevated level of HNE in blood plasma and cells of the animal and human body is observed in the course of many diseases, including cancer. On the other hand, it is currently proven that cancer cells divert to apoptosis if they are exposed to supraphysiological levels of HNE in the cancer microenvironment. In this review, we briefly summarize the current knowledge about the biological properties of HNE.

## 1. Introduction

The interest in aldehydes as secondary products of lipid peroxidation began with the intensive research of Esterbauer and his colleagues conducted in the 1970s and 1980s [1,2,3,4,5]. HNE is an extraordinarily reactive compound since it possesses three functional groups: (1) an aldehyde (carbonyl) group which can yield acetal/thioacetal or can be a target for Schiff-base formation, oxidation, or reduction; (2) a double bond between carbon C2 and C3 that can be a target for Michael additions to form a thiol or can undergo reduction or epoxidation, and (3) a secondary alcohol group at carbon C4 which can be oxidized to a ketone and which facilitates C=C polarization and cyclization reactions [6]. Thus, carbon C1 and C3 are electrophilic sites and carbon C1 is also a redox center. Generally, HNE is considered to be a soft electrophile (Figure 1)

The research on aldehydes conducted several decades ago by Esterbauer and his colleagues led researchers to conclude that malondialdehyde (MDA) appeared to be the most mutagenic product of lipid peroxidation, whereas HNE was the most toxic one [7]. In general, for many years these reactive aldehydes defined as the “thiobarbituric acid reactive substances” (TBARS) were considered mostly as toxic end-products of oxidative degradation of lipids, mainly polyunsaturated fatty acids (PUFAs; LH).

A review of the current publications has shown that HNE is still referred to as a toxic molecule, but nevertheless studies also increasingly show the positive aspects of its biological activity. In an almost poetic way, HNE can be said to behave similar to “Dr. Jekyll and Mr. Hyde”. Schaur points out that for the biological effects of HNE it is important that its lipophilic properties prevail over the hydrophilic ones and, therefore, HNE tends to concentrate in biomembranes rather than in the aqueous phase of cells [8]. 

In this review, we briefly summarize the current knowledge about the biological properties of HNE. 

## 2. The Formation and Removal of HNE

As mentioned above, HNE is formed in the reaction of lipid peroxidation which is the best-known biological process of all processes that proceed according to a free radical chain reaction scheme. It is well known that radical chain reactions have three phases: initiation, propagation, and termination. Initiation involves the free oxygen radical-induced abstraction of a hydrogen atom from the free LH or the rest of such acids that are a part of phospholipids. Usually, one of the hydrogen atoms bound to the carbon atom situated between the two double bonds is abstracted, because it is known that the double bonds weaken the carbon-hydrogen bond in the adjacent carbon atoms. At this step, a fatty acid radical (L^•^) is produced. It is a reactive molecule to which molecular oxygen (O_2_) is added in the next step of the radical substitution to generate a lipid peroxyl radical (LOO^•^), which then triggers a chain reaction. What factors can abstract the hydrogen from the PUFA molecule? There are numerous initiators of lipid peroxidation in biological systems—often they are reactive oxygen species (ROS), such as hydroxyl radical (^•^OH), ozone (O_3_), nitrogen oxide (NO) and dioxide (NO_2_), and sulfur dioxide [9]. Contrary to the original opinions from almost half a century ago, the superoxide anion (O_2_**^•−^**) is not a factor capable of initiating the lipid peroxidation process. On the other hand, the protonated form of the O_2_**^•−^**, i.e., the hydroperoxide radical (HO_2_^•^) is able to initiate the lipid peroxidation process. Taken together, as a result of the initiation reaction, the PUFA molecule is converted into an L^•^, that is, an unpaired electron remains at the carbon atom from which the hydrogen atom has been abstracted. In the molecule of L^•^ formed in this way, rearrangement of double bonds results in the formation of conjugated dienes, i.e., molecules that have the conjugated double bonds separated by one single bond. The conjugated dienes are more stable than other dienes because of resonance. The conjugated dienes are characterized by a light absorption peak at 234 nm. The measurement of an increase in absorbance at this wavelength is, therefore, a convenient way to monitor the lipid peroxidation process [10,11]. 

Propagation is the step consisting in the reaction of L^•^ with O_2_, resulting in the formation of lipoperoxyl radical (LOO^•^), which can now transfer its radical function through hydrogen abstraction to other unsaturated fatty acids, which in turn add O_2_ and thus keep the chain reaction going. This reaction yielding a new L^•^ and a lipid hydroperoxide (LOOHs), can be expressed as follows: LOO^•^ + LH → LOOH + L^•^


The LOO^•^ can also react with a molecule with antioxidant properties (e.g., α-tocopherol or γ-tocopherol). In this case, the chain reaction is interrupted. The radical chain reaction usually is in full swing once all antioxidants have been completely consumed (lag time). Thus, one hydroxyl radical can generate a large number of lipid hydroperoxides until the chain reaction is terminated by a chain-breaking antioxidant.

At this point, it is also worth mentioning the phenomenon of reinitiation, consisting in the breakdown of LOOH under the influence of transition metal ions, such as Fe^2+^, Fe^3+^ and Cu^2+^, which leads to the generation of free radicals: LO^•^ and LOO^•^ [12]. It is also known that glucose and glycosylated proteins and peptides may participate in the reinitiation process [13]. 

The process of lipid peroxidation can be described schematically as follows:*Initiation*:LH + ^•^OH → L^•^ + H_2_O*Propagation*:L^•^ + O_2_ → LOO^•^*Propagation*:LOO^•^ + LH → LOOH + L^•^*Propagation*:LOOH → LO^•^, LOO^•^, ^•^OH, aldehydes, ketones, alcohols*Termination*:L^•^ + L^•^ → L-L (lipid dimer; stable nonradical product)*Termination*:LOO^•^ + LOO^•^ → LOOL (stable nonradical product) + O_2_*Termination*:L^•^ + LOO^•^ → LOOL (stable nonradical product)

The primary products of lipid peroxidation, namely LOOH molecules are very unstable, therefore, they can be transformed into peroxyl and alkoxyl (LO) radicals and can be decomposed to secondary products. It is known that carbon-carbon (C-C) bond located at the β-position of an alkoxy radical readily breaks to form an alkyl radical along with a carbonyl compound. Such processes are called β-scission or β-fragmentation [14,15]. The β-fragmentation process results in the formation of short-chain products, including HNE. A schematic representation of the HNE formation is illustrated by the process of arachidonic acid peroxidation in Figure 2.

The process described above takes place in a non-enzymatic manner. Some authors also describe the production of HNE and other aldehydes by enzyme-catalyzed lipid peroxidation [17]. Additionally, ω-6 PUFAs, such as linoleic and arachidonic acid are substrates for the production of HNE by an enzymatic process. The main enzymes involved in this process include 15-lipoxygenases (15-LOX): 15-LOX-1 and 15-LOX-2, which, unlike other LOXs, can act not only on free fatty acids released from phospholipids under the influence of phospholipases but also on the whole lipid molecules in cell membranes. The 15-LOX-1 is expressed in reticulocytes, eosinophils, and macrophages. The 15-LOX-2 is expressed in the skin, cornea, prostate, lung, and esophagus [18,19]. The main precursors of HNE in humans comprise 13-hydroperoxyoctadecadienoic acid (13-HPODE) produced by the oxidation of linoleic acid catalyzed by 15-LOX-1 and 15- hydroperoxyeicosatetraenoic acids (15-HPETE) produced by the oxidation of arachidonic acid catalyzed by 15-LOX-2 [17,20].

Lipids also can be oxidized by cyclooxygenases (COXs). However, it should be remembered that as a result of the action of LOXs and COXs on PUFAs, molecules called eicosanoids are formed. There are multiple subfamilies of eicosanoids, including the most prominent leukotrienes, lipoxins, resolvins, eoxins and prostaglandins, and thromboxanes. Eicosanoids are intra- and extracellular signaling substances that can be produced by any cell and usually act through G protein-coupled membrane receptors. Eicosanoids are involved in the regulation of inflammatory responses, such as fever and allergies, and also play an important role in regulating the cardiovascular system, blood pressure, salt secretion, blood clotting and pain. Therefore, they are compounds with an extremely wide range of actions in both physiological and pathological conditions. Thus, in our opinion, the production of lipid-free radicals and HNE is a side effect of LOX and COX action. On the other hand, Wang et al. demonstrated that *Enterococcus faecalis* (*E. faecalis*), a human intestinal commensal bacteria, can stimulate macrophages to produce HNE through COX-2 [21]. The concentration of HNE under physiological conditions is below 1 μM, and this molecule displays a half-life of less than 2 min because it can be detoxified in various reactions of biotransformation or it can form macromolecular adducts [22]. The phrase “various reactions of biotransformation” means enzymatically catalyzed detoxification reactions of HNE. The phrase “or it can form macromolecular adducts” means that the main purpose of this very rapid removal of HNE by enzymatically catalyzed reactions is the defense against modification of macromolecules (proteins, lipids and nucleic acids) by the extremely reactive aldehyde group of HNE, that is the formation (production) of “macromolecular adducts”. In a word, when enzymatically catalyzed processes of HNE removal fail, this compound forms adducts with macromolecules. 

For the first time, enzymatic detoxification of HNE was described by Ferro et al. in 1988 [23]. Current knowledge on this subject is already quite large.

There are three enzymatic pathways of HNE metabolism: (1) reduction, resulting in the corresponding alcohol, namely 1,4-dihydroxy-2-nonene (DHN). This reaction is catalyzed by alcohol dehydrogenases (ADHs); (2) oxidation, yielding the acid, namely 4-hydroxy-2-nonenoic acid (HNA). This reaction is catalyzed by aldehyde dehydrogenases (ALDHs); (3) formation of a conjugate with the reduced glutathione (GSH) catalyzed by glutathione-S-transferases (GSTs). These conjugates undergo further transformations and are finally excreted from the body in the urine as mercapturic acid (MA). Therefore, HNE is cleared from the body, but at the same time the cellular pool of cysteine, which is a strong antioxidant and substrate for the synthesis of GSH, is depleted, which additionally impairs the antioxidant defense of the cell [24]. Doorn and Petersen showed that the reaction between HNE and GSH could also occur spontaneously, and such spontaneous conjugation was found to occur with a constant rate of 1.33 M^−1^ s^−1^ for HNE [25]. Essentially, however, the non-enzymatic routes of HNE metabolism lead to the formation of the adducts with macromolecules.

### 2.1. Protein Adducts

HNE easily reacts with nucleophilic groups of proteins. It is a type of non-enzymatic, covalent modification that leads to the formation of adducts called advanced lipoxidation end products (ALE). They are either Michael products or Schiff’s bases [26,27,28,29]. 

The Michael reaction (Michael addition) is the nucleophilic addition of a nucleophile to an α,β-unsaturated carbonyl compound containing an electron-withdrawing group. The reaction was named after the American chemist Arthur Michael (1853–1942), who first published it in 1887 [30,31]. As a result of this reaction, single carbon-carbon, and carbon-sulfur, carbon-oxygen and carbon-nitrogen bonds can be formed. The, α, β-unsaturated HNE is susceptible to nucleophilic attack from cysteine (Cys), histidine (His) and lysine (Lys) residues present in the polypeptide chain (Figure 3). Doorn and Petersen demonstrated that HNE reacted with amino acid nucleophiles via Michael addition with the following order of potency: Cys ≫ His > Lys and indicated that Arg was not a target for HNE. These authors also showed that the presence of an Arg on a Cys-containing peptide increased the reaction rate with HNE by a factor of approximately 5–6 compared to the Cys nucleophile alone [25]. 

It is worth mentioning that Sayre et al. found mass spectrometric evidence for a trace Arg-HNE adduct that can be reductively trapped as a mixture of Michael and cyclic Schiff base-Michael adducts. However, the authors were not able to isolate any stable adducts of Arg models with HNE under the conditions of the conducted experiment. The authors also emphasize that the physiological significance of this HNE-derived Arg adduct is unclear [29]. 

The Schiff bases are named after Hugo Schiff (1834–1915), the German-Italian chemist, and discoverer of this class of substances [32]. These substances are the condensation products of primary (1°) amines and carbonyl compounds. They can be considered a sub-class of imines. So, HNE is involved in the formation of Schiff bases by the reaction of its aldehyde group with Lys and residues present in the polypeptide chain.

The formation of both Michael products and Schiff bases results in changes in the protein structure, including the production of intra- and intermolecular bonds that affect the cross-linking of proteins. Many studies have shown that adducts of this type are toxic, and their accumulation in the cell is closely related to the development of many diseases. In recent years, a lot of reviews have recapitulated the relationship between the presence of HNE-protein reaction products in the body and the etiopathogenesis of many disorders and diseases (the onset of the disease or disorder and also its progression and maintenance). So, elevated levels of oxidatively modified proteins have been linked to diverse cardiovascular diseases [33], liver inflammation [34], renal failure [35] and neurodegenerative disorders [36], as well as aging [37,38]. 

Readers who are more deeply interested in a detailed discussion of these problems may to a number of very interesting and very good reviews [6,8,14,24,39,40,41,42,43,44,45]. 

Given the ability of the carbonyl group to react with alcohols and thiols, the products of which are hemi- and thiohemiacetals, it is only a matter of respectively to conduct an experiment showing that LPO-derived aldehydes, including HNEs, form adducts with proteins forming hemi- and thiohemiacetals with serine (Ser), tyrosine (Tyr), threonine (Thr) and Cys, respectively residues present in the polypeptide chain. From a basic organic chemistry course, it is known that aldehydes also react with alcohol and thiol groups to form hemi- and thiohemiacetals, respectively. So, there is a likelihood that HNE (and other LPO-derived aldehydes) could react with such amino acid residues in a polypeptide chain as serine (Ser), tyrosine (Tyr), threonine (Thr) and Cys, forming hemi- and thiohemiacetals, respectively. There is no such evidence yet, but in our opinion, this thesis seems very plausible. 

The existence of this type of reaction of proteins with other aldehydes (e.g., glyoxal) has been shown, however, there is no such data for LPO-derived aldehydes, including HNE [46]. This proposition is all the more plausible, since the existence of intramolecular hemiacetals formed by such Michael adducts, which are formed by the reaction of HNE with thiol compounds, for example with cysteine present in proteins, has already been demonstrated [8]. Figure 4 (scheme) below shows a plausible mechanism of the formation of the hemi- and thiohemiacetals in the reaction of HNE with Ser, Tyr, Thr and Cys residues present in the polypeptide chain. The scheme shows also the formation of hemiacetal with a Michael adduct of HNE.

### 2.2. Lipid Adducts

Until the early 1990s, there were no data available describing HNE interactions with lipids. Only in 1998, Guichardant et al. were the first to report in vitro conditions for the covalent reaction between HNE and aminophospholipids, such as phosphatidylethanolamine (PE) and phosphatidylserine (PS) (Figure 5).

It was then shown that PE reacted with HNE to form the same kind of products as with amino acids in proteins, namely a Michael adduct that represents the main derivative and a minor Schiff base adduct, which was partly cyclized as a pyrrole derivative via a loss of water. By contrast, PS was found to poorly react with HNE, producing only a small amount of Michael adduct while the Schiff-base was undetectable [47]. 

Bacot et al. for the first time identified Michael adducts formed between HNE and PE in biological membranes. These authors also showed that the PE-HNE concentrations of 10 and 25 μM potentiated the aggregation of human blood platelets, while, on the contrary, 200 μM PE-4HNE Michael adducts, inhibited the human blood platelet aggregation by about 70% [48]. The formation of the Michael adducts and Schiff bases in the reaction of PE with HNE was also found in a study by Vazdar et al. Those authors identified a cyclization of initially formed Michael adducts to hemiacetals, whereas in the case of Schiff base products, the authors isolated the product mixture of the Schiff base and the pyrrole derivative. The authors emphasize that the formation of the pyrrole derivative of an initially formed Schiff base is strongly thermodynamically favored [49]. 

### 2.3. Adducts with Nucleic Acids

It is known that HNE is much more reactive to proteins than to DNA. HNE is genotoxic mainly for hepatocytes and cerebral endothelial cells [6]. It is now known that HNE undergoes oxidative conversion to an epoxy derivative (epox-HNE; EH). Compared to HNE, EH is more reactive and readily modifies nucleobases to form ethenoadducts. It primarily reacts with deoxyadenosine (dA) and deoxyguanosine forming unsubstituted and substituted etheno purine adducts, such as: 1,N_6_-ethenodeoxyadenosine (εdA), 1,N^2^-ethenodeoxyguanosine (εdG), 7-(1′,2′-dihydroxyheptyl)-1,N^2^-ethenodeoxyguanosine (DHHεdG), and 7-(1′,2′-dihydroxyheptyl)-1,N^6^-ethenodeoxyadenosine (DHHεdA). It should be remembered that the EH exists as a pair of diastereomers and each of its diastereomers exists as a pair of enantiomers. So, it is not difficult to see that two pairs of diastereomers are formed in the reaction of EH with dA because of the two chiral carbons in the side chain [50]. 

The derivative DHHεdA has been extensively investigated in recent years because it is believed to play an important role in carcinogenesis. Admittedly, HNE-dG has also been shown to be mutagenic and EH also modifies dG to yield DHHεdG, but the level of dG adducts in vivo are relatively low, possibly due to their poor stability [51,52]. Hence, the derivative DHHεdA has become a major research target in recent years. Dyba et al. described the development of monoclonal antibodies specifically against DHHεdA and proposed their application to detect DHHεdA in human cells [53]. Additionally noteworthy are the results of the research showing for the first time the elevated DHHεdA levels in almost 81% of liver biopsies from young non-alcoholic steatohepatitis (NASH) patients [54]. It is worth mentioning at the end of this thread that Feng et. al. have shown that HNE not only modifies nucleobases to form ethenoadducts, but also inhibits DNA repair by direct modification of DNA repair proteins in vitro. The authors have also found that HNE can greatly enhance the sensitivity of human cells to benzo[a]pyrene diol epoxide (BPDE), which is a major carcinogen in cigarette smoke and environment and enhances the UV-induced human colon and lung epithelial cell killing [55]. Thus, the inhibition of DNA repair by HNE and possibly other LPO-derived aldehydes can be seen as an important cause of many human diseases, including cancer.

## 3. HNE in Pathology

HNE is a physiological component of the human and animal bodies. The normal HNE concentrations of blood serum in healthy adults and children (except neonates) were found to be in the range from 0.05 to 0.15 μM [56]. An increase in the level of HNE in the blood and cells is often accompanied by a simultaneous increase in the level of malondialdehyde (MDA) and other thiobarbituric acid reactive substances (TBA-RS), as well as by a decrease in the level of GSH and an increase in the level of disulfides. These types of changes have been observed in tissues and cells in the course of many diseases, such as neurodegenerative diseases, cardiovascular and pulmonary diseases, metabolic syndrome, visual disorders, gastroesophageal reflux disease, chronic inflammatory diseases, immune disorders and aging. A lot of experimental and review papers have been published about the role of HNE and its adducts with proteins, lipids and nucleic acids in the course of these diseases, some of which are also cited in this publication. Readers who are more interested in this topic, please refer to a very interesting and very good review by Schaur and colleagues, also quoted in the present publication [6]. The role of HNE in the development of neoplastic cells was mentioned above. Besides, the evidence suggesting a role for HNE in the pathogenesis of many types of cancer also has been reviewed in many papers [57,58,59,60,61]. It is known that in pathological conditions elevated levels of HNE-modified molecules are found in the blood and other tissues. Some HNE-modified molecules, which can contribute to the pathophysiology of various human disorders are shown in the table below (Table 1).

At this point, however, we would like to draw the readers’ attention to several recent papers addressing less commonly known aspects of the pathological role of HNE in the human and animal body.

Current knowledge allows us to regard HNE as a pathophysiological factor and biomarker of numerous stress-associated diseases, as well as mental and psychosomatic disorders. Rosen et al. conducted a study involving depressed coronary artery disease (CAD) patients. The obtained results indicated that serum HNE concentrations significantly increased in depressed CAD patients compared to non-depressed CAD patients. In these studies, the authors used the Structured Clinical Interview for DSM Axis I Disorders—Depression Module (SCID) to diagnose depression at baseline and the Center for Epidemiological Studies Depression Scale (CES-D) to measure depressive symptoms. It turned out that serum HNE decreased less in CAD patients with depression compared to non-depressed subjects. In addition, a decrease in HNE concentrations was significantly associated with a decrease in CES-D scores over 6 months. In the authors’ opinion, the results obtained by them suggest that HNE may be an important marker of depressive symptoms in CAD and may be involved in its progression [69]. Studies from Perković et al. conducted in war veterans with posttraumatic stress disorder (PTSD) revealed significantly elevated levels of plasma HNE in these patients. Moreover, the authors have shown that accumulation of plasma HNE seems to increase with aging, but it is negatively correlated with BMI, showing a specific pattern of changes for individuals diagnosed with PTSD [70]. There are also interesting studies in young patients (mean age 12.0 ± 6.2 years) with classic autism in which the levels of non-protein-bound iron (NPBI), a pro-oxidant factor, and HNE protein adducts (HNE-PAs) were examined for the first time. The results obtained by the authors indicated that PA-HNE was significantly higher in the erythrocytic membranes and plasma of autistic patients compared to the control group. Moreover, the authors observed an increased level of the pro-oxidant factor, non-protein-bound iron (NPBI). Therefore, it seems that NPBI may contribute to lipid peroxidation and, consequently, to increased plasma and erythrocytic membrane HNE-PAs thus amplifying the oxidative damage and potentially contributing to the autistic phenotype [71]. 

To conclude this topic, we would like to draw the readers’ attention to the interconnection between eating disorders and oxidative stress. The most famous eating disorder is anorexia. The systematic review and meta-analysis of oxidative stress and antioxidant markers in anorexia nervosa (AN) by Solmi et al. revealed that compared to healthy controls, AN patients showed significantly higher levels of nitric oxide (NO)-related parameters (platelet NO, exhaled NO and nitrites) and significantly lower GSH and free Cys levels [72]. Although the authors did not analyze the level of lipid peroxidation products, including HNE in AN patients, these studies clearly show that AN is associated with some markers of increased oxidative stress, including those associated with increased levels of HNE, such as lower GSH and free Cys levels. In addition, Zimniak proposed a simple and, from the point of view of biochemistry, an elegant concept of a pro-obesity mechanism, based on the known fact that HNE inhibits the activity of aconitase, one of the enzymes of the Krebs cycle. The inhibition of aconitase causes an accumulation of its substrate, citrate. Citrate excess may be transported from mitochondria to fulfill a dual function: supplying the substrate for acetyl-CoA carboxylase (ACC) and its allosterical activation. ACC catalyzes the reaction to form malonyl-Coa with acetyl-CoA. It is the first, key reaction of the fatty acid synthesis pathway and, consequently, of fats [73].

### A Dual Role of HNE in Cancer

Apoptosis is recognized as the main mechanism in preventing cancer. It is worth remembering that most (and perhaps even all) types of cancer cells are insensitive to apoptosis. In this context, the results of the research by Biasi et al. are interesting. These authors investigated the crosstalk between SMADs (a family of proteins that play the role of transcription factors, involved in signal transduction induced by transforming growth factor beta TGF-β) and c-Jun NH2-terminal kinase (JNK) signal transduction pathways in inducing apoptosis. For this purpose, these authors added HNE and TGF-β1 (singly or in combination) to CaCo-2 human colon adenocarcinoma cells. The obtained results indicated that the co-treatment induced a marked enhancement of apoptosis, much greater than either individual molecule. However, in the presence of a JNK inhibitor, the cooperative proapoptotic effect was abolished. It means that in cancer cells that are resistant to growth inhibition by TGF-β1, the addition of HNE, for which JNK is the signaling target, may be helpful in inhibiting tumor growth [74]. It is known that the TGF-β1/SMAD signaling pathway plays also an important role in chronic myeloid leukemia cells and leads to growth inhibition, and apoptosis of the leukemia cells [75]. Some authors suggest that HNE can induce signaling for apoptosis via multiple pathways, which seem to converge in the activation of JNK and caspase-3 [76]. 

On the other hand, the serine/threonine protein kinase Akt also called protein kinase B (PKB), is a major transducer of the phosphatidylinositol 3-kinase (PI3K) pathway and plays a crucial role in the regulation of cellular processes, such as survival and proliferation. The increased expression and activation of Akt is found in many human cancers. Basically, the Akt is considered a fundamental player in the process of cancer advancement; its elevated activity is generally correlated with tumor progression and poor prognosis [77]. Some authors indicated that HNE at lower concentrations (as low as 0.1 μM) triggered phosphorylation of epidermal growth factor receptor (EGFR) and activation of its downstream signaling components, including Akt, which led to increased cell proliferation [78]. So, it is obvious that HNE-mediated regulation of signaling pathways is pleiotropic and the impact of HNE on signaling pathways depends on the concentration of HNE. Usually, in cell culture studies, HNE concentration greater than 10 μM has been reported to induce apoptosis whereas sub-lethal dose ≤5 μM induces cell proliferation [45]. 

It should be emphasized that free HNE remains at very low level in plasma, cells, and tissues under physiological conditions. In human plasma, it is in the range of 0.28–0.68 μM [79]. In other types of tissues, HNE concentration is higher than in plasma. For example, the HNE level in human monocytes is about 10 times higher than in human plasma [80]. However, due to a tissue-dependent HNE metabolizing capacity, as already mentioned, HNE concentration in different tissues may vary under physiological conditions [80]. The observations of Doorn et al. showing that HNE is both a substrate and an inhibitor of mitochondrial aldehyde dehydrogenase (ALDH2) are also interesting. These authors indicated that ALDH2 inhibition is reversible at a low concentration of HNE and becomes irreversible when the concentration of HNE reaches10 µM [81]. Some studies indicated that HNE-induced ALDH2 inactivation may have an important role in the progression of some cancers [82].

It is commonly known that there is a mutual relationship between the altered redox status in cancer cells and the increase in lipid peroxidation induced by ROS and the subsequent formation of reactive lipid electrophiles, including HNE. Many experiments have indicated that tumor cells “manipulate” their redox status to acquire anti-apoptotic phenotype. Authors who have been studying this issue in recent years drew attention to cardiolipin (IUPAC name 1,3-bis (sn-3’-phosphatidyl) -sn-glycerol; CL). CLs are a structurally unique class of phospholipids—more specifically glycerophospholipids—because the CL molecule is a dimer composed of two phosphatidyl residues linked by a glycerol bridge. In other words, each molecule of CL consists of two fatty acids attached in an ester linkage to the first and second carbon of glycerol, and glycerol attached through a phosphodiester linkage to the third carbon of the glycerol, to which the second molecule of glycerol is attached through a phosphodiester linkage, having two fatty acids attached in ester linkage to its first and second carbon. Thus, each CL molecule has four acyl chains and three glycerol backbones in the same molecule, while glycerophospholipids commonly have two fatty acids and one glycerol unit (Figure 6).

CL is an important component of the inner mitochondrial membrane, where it constitutes about 20% of the total lipid composition. The studies of Liu et al. provided the first evidence in vitro and in vivo for the formation HNE from CL oxidation, especially from tetralinoleoyl cardiolipin (L_4_CL), via cross-chain peroxyl radical addition and decomposition, which may have implications for apoptosis and other biological activities of HNE [83]. Kagan et al. indicated that CL was the only mitochondrial phospholipid that underwent early oxidation during apoptosis. Oxidation is catalyzed by the CL-specific cytochrome c peroxidase activity. The authors also showed that oxidized CL was required for the release of pro-apoptotic factors. These results provide insight into the role of reactive oxygen species in triggering the cell death pathway and describe the early role of cytochrome c before caspase activation [84]. It should be remembered that programmed cell death (apoptosis) is desirable by the organism. It is a process that ensures the safety of the body because apoptosis is involved in removal of damaged or unwanted cells to ensure their clean and safe self-elimination. A potential relationship between mitochondrial CL oxidation, generation of lipid electrophiles, including HNE, mitochondrial dysfunction and apoptosis during the progression of atherosclerosis is also indicated by studies reported by Zhong et al. [85]. 

Based on the above-mentioned facts, an irresistible conclusion can be drawn that HNE has a dual role in cancer. The conclusion is correct, and it is confirmed by numerous scientific studies [43]. Zhong et. al. demonstrated that a high-PUFA diet i.e., rich in linoleic, docosahexaenoic arachidonic acid could serve as an effective adjuvant therapy in the treatment of cancers by a mechanism linking apoptotic cell death to mitochondrial HNE formation, oxidative stress, and lipid peroxidation. These authors demonstrated also that Sorafenib-induced cell death correlated with the levels of L_4_CL and its oxidation, suggesting that incorporation of linoleic acid into CL of cancer cells may sensitize the cancer cells to killing by standard treatment with Sorafenib in hepatocellular carcinoma (HCC) [68]. 

Therefore, attention is increasingly drawn to the fact that HNE—the most studied final lipid peroxidation product—is a promising new molecule in anticancer therapeutic strategies (sic!) [86]. It was also indicated that the inclusion complex of HNE with a polymeric derivative of β-cyclodextrin enhanced the anticancer efficacy of the aldehyde in several tumor cell lines and in a three-dimensional human melanoma model [87].

Thus, HNE, considered to be an important factor in carcinogenesis due to its ability to covalently bind to DNA, may also be cytotoxic to cancer cells as well as may modulate their growth. In addition, as a brief review shows, HNE also contributes to the mechanisms which enhance the effects of cytostatic drugs thus increasing the effectiveness of radiotherapy [24]. It is also known that some neoplastic cells may undergo apoptosis or necrosis if exposed to supraphysiological levels of HNE in the cancer microenvironment, especially if they are additionally provoked by pro-oxidative cytostatics and/or inflammation [24]. These findings may explain the previously observed disappearance of HNE from invading cancer cells, which is associated with an increase in HNE content in benign cells close to the invading cancer, which was investigated using CL as a source of tumor-suppressing HNE. In addition, anticancer treatments may target the metabolic pathway for HNE detoxification through RLIP76—a multifunctional membrane protein that transports glutathione conjugates of electrophilic compounds and other xenobiotics, including chemotherapeutic agents, out of cells [24]. 

## 4. Conclusions

### HNE Is Not Only Harmful but Also Beneficial

HNE started its “career” in the 1990s as a “toxic product of lipid peroxidation” and the “second toxic messenger of free radicals”. However, already at the beginning of the 21st century, HNE was perceived as a reliable marker of oxidative stress, growth modulating factor and signaling molecule. Currently, it is proven that cancer cells divert to apoptosis if they are exposed to supraphysiological levels of HNE in the cancer microenvironment. An increasing body of data indicates that the primary dysfunction of the tumor microenvironment is of key importance for the process of carcinogenesis. The key aspects of the carcinogenic process include oxidative stress and inflammation. The anticancer activity of HNE through interference with the inflammatory aspects of carcinogenesis has been revealed only recently and is currently the subject of very intensive research. 

In general, the current view is that higher HNE levels in cells promote apoptotic signaling while at mild levels below its basal constitutive levels, 4HNE promotes proliferation [87,88].

In the opinion of Zarkovic, “it is likely that HNE will soon become one of the most attractive factors for those who search for a small and reactive molecular link between genomics and proteomics” [89]. 

## Figures and Tables

**Figure 1 biomolecules-12-00145-f001:**
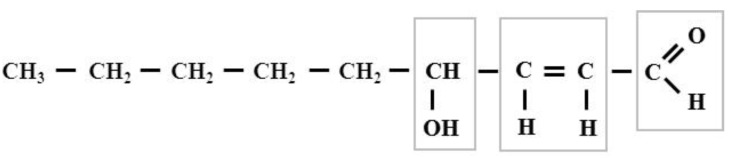
The formula of 4-hydroxy-2,3-trans-nonenal. The squares indicate the reactive groups of this compound. The cis-aldehydes are not stable, they are easily cyclized to form.

**Figure 2 biomolecules-12-00145-f002:**
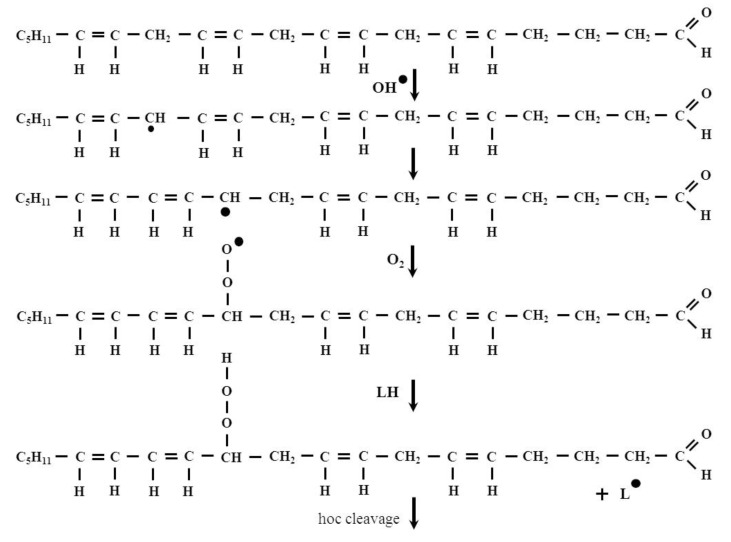
The formation of HNE in the arachidonic acid peroxidation process. 2-Nonenal is one of the intermediates of this process. It is known that the odor of this compounds is perceived as orris, fat and cucumber and it has been associated with human body odor alterations during aging [16].

**Figure 3 biomolecules-12-00145-f003:**
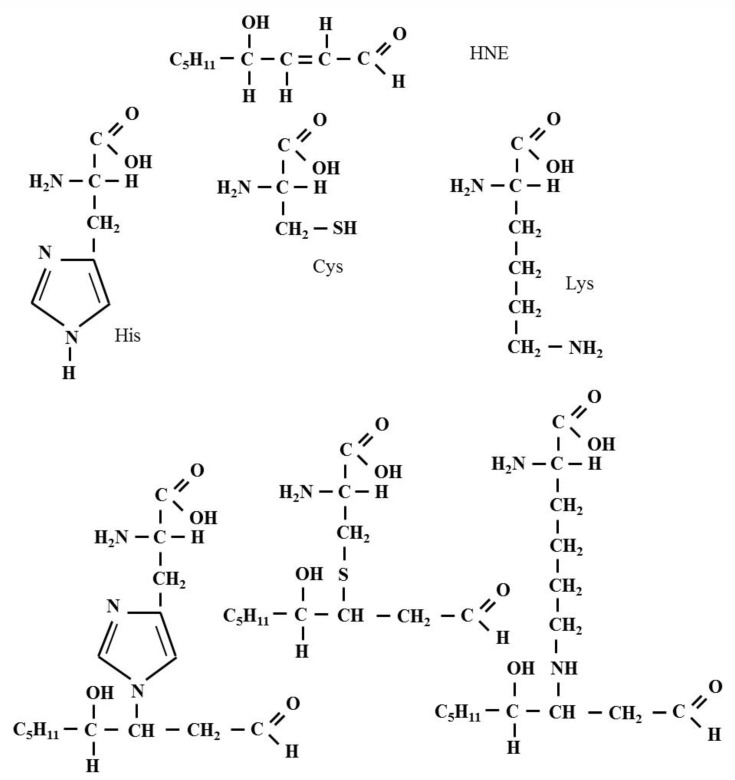
The formation of the Michael adduct in the reaction of HNE with Cys, His and Lys residues present in the polypeptide chain. As a result of this reaction, chemical bonds are formed, including a single carbon-sulfur bond and a single carbon-nitrogen bond.

**Figure 4 biomolecules-12-00145-f004:**
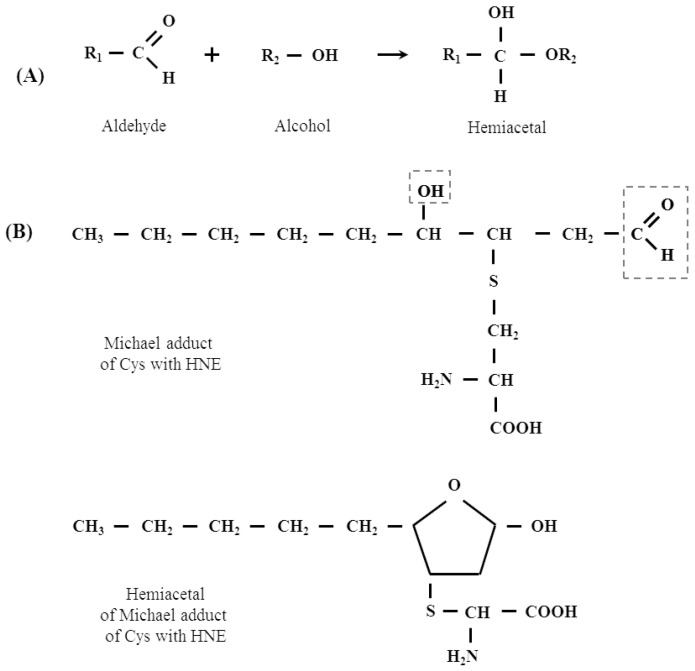
(**A**) The mechanism of hemiacetal formation in the reaction of an alcohol with an aldehyde. (**B**) Hemiacetal formation from the reaction of Michael adduct of Cys with HNE. (**C**) A plausible mechanism of the formation of the hemi- and thiohemiacetals in the reaction of HNE with Ser, Tyr, Thr and Cys residues present in the polypeptide chain.

**Figure 5 biomolecules-12-00145-f005:**
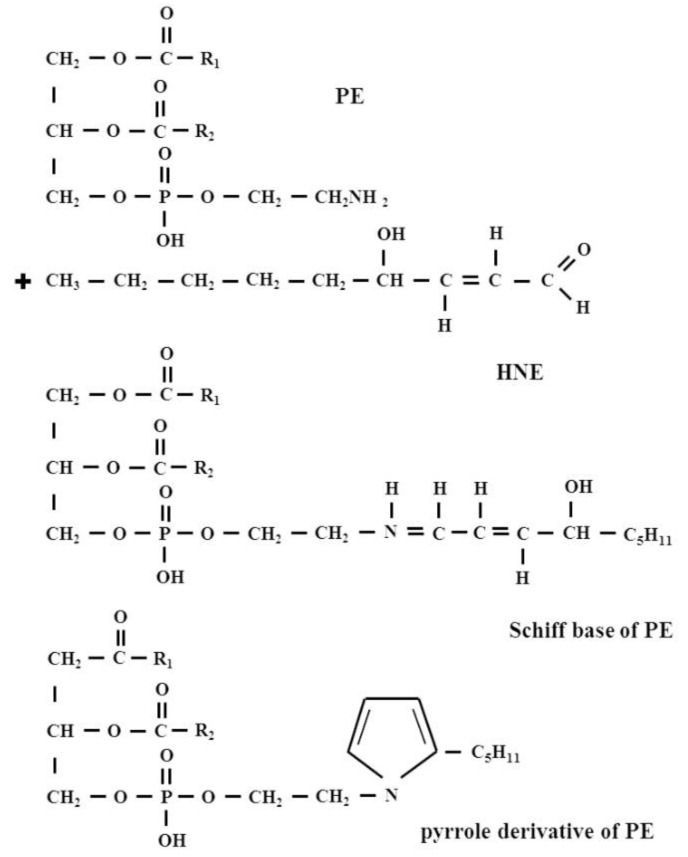
The formation of HNE-PE adducts: a Michael adduct and Schiff base adduct and also Schiff base adduct, which cyclizes as a pyrrole derivative via a loss of water.

**Figure 6 biomolecules-12-00145-f006:**
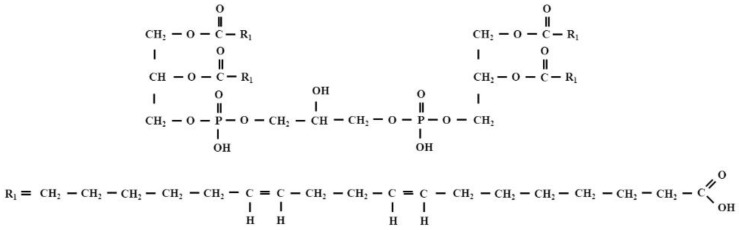
The structural formula of tetralinoleoyl cardiolipin (L_4_CL).

**Table 1 biomolecules-12-00145-t001:** Importance of some HNE-modified proteins for practical clinical medicine.

HNE-Modified Molecules	Mode of Action	Medical Aspects	Some References
Activator protein 1 (AP-1) transcription factor	HNE alters AP-1 transcriptional activity	Vascular complications	[62]
Nuclear factor kappa-light-chain-enhancer of activated B cells (NF-*κ*B)	HNE can induce either activation or inhibition of NF-*κ*B	Increase the NF-*κ*B activity is nvolved in inflammatory signaling in rheumatoid arthritis	[45,63]
A non-receptor protein tyrosine kinases family; Src kinases	HNE induces the activation and phosphorylation of these kinases Activation of Src by HNE leads to the expression of inflammatory mediator Cox2 and transcription factor AP-1, via activation of p38MAPK, JNK and ERK1/2	The HNE-Src adduct is pro-inflammatory in aged kidneys	[64,65]
Peroxisome proliferator-activated receptors delta (PPARδ)	HNE and its derivatives are the endogenous ligands for PPARδ	HNE and its derivatives as ligands of PPAR are proposed as drugs (or they serve as prototype molecules for the development of such therapeutic agents) in the treatment of metabolic syndrome diseases. Metabolic syndrome is associated with the risk of developing cardiovascular disease and type 2 diabetes.	[20]
Adipose proteins	HNE can form adduct with these proteins	Insulin resistant obesity	[66]
Human serum albumin (HSA)	HNE can form adduct with HSA, making it highly immunogenic	Systemic lupus erythematosus (SLE), a chronic autoimmune disease,	[67]
Apolipoprotein B (ApoB)	HNE can form adduct with ApoB	Atherosclerosis; increasing the risk of athero-thrombotic events.	[68]

## Data Availability

Not applicable.

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
