# Peer review of "Chemistry and Biochemistry Aspects of the 4-Hydroxy-2,3-trans-nonenal"

_biomolecules, 2022, doi:10.3390/biom12010145_

Round 1
Reviewer 1 Report
134 ... Missing space: ... formed.There ...
141... As there are 3 authors, instead of "in my opinion" write "in OUR opinion."
212 ... analogy to 141: Use plural insrtead of singular
218 ... not understandable sentence: Given ... it is only a matter of RESPECTIVELY to conduct ... Do you mean "...it is only a matter of analoguous experimental setting..."
223 ... use plural: " ... no such data for LPO-derived aldehydeS, such as HNE."
399 ... Missing citation after "...promising new molecule in anticancer therapeutic strategies (SIC!)" If you copy a statement you should indicate its origin.
406 ... Missing citation after "... radiotherapy."
408 ... Missing citation after " ... inflammation."
543 ... Journal name is eradicated!
Author Response
Reply to Reviewer 1
We are grateful to the Reviewer for valuable comments and thorough review of our paper.
Please, find my replies below.
134 ... Missing space: ... formed.There ...
It has been corrected.
141... As there are 3 authors, instead of "in my opinion" write "in OUR opinion."
It has been corrected.
212 ... analogy to 141: Use plural insrtead of singular
It has been corrected.
218 ... not understandable sentence: Given ... it is only a matter of RESPECTIVELY to conduct ... Do you mean "...it is only a matter of analoguous experimental setting..."
The Reviewer is right to point out that this sentence is not very clearly written. This sentence has been reworded.
223 ... use plural: " ... no such data for LPO-derived aldehydeS, such as HNE."
It has been corrected.
399 ... Missing citation after "...promising new molecule in anticancer therapeutic strategies (SIC!)" If you copy a statement you should indicate its origin.
It has been corrected.
406 ... Missing citation after "... radiotherapy."
It has been corrected.
408 ... Missing citation after " ... inflammation."
It has been corrected.
543 ... Journal name is eradicated!
It has been corrected.
Taking into account comments of the Reviewer suggesting that English needs improvement we want to inform you that that our paper was checked for proper English usage by a professional Polish-English translator in the field of scientific papers.

Reviewer 2 Report
Comments to the Author
General comments
In this manuscript, Bilska-Wilkosz et al. reviewed and reported a perspective view on physiological functions of 4-HNE. They attentively explained the formation and removal of 4-HNE. In addition, they collected a lot of information about the role of 4-HNE in pathology. However, they expounded less about molecular mechanisms of 4-HNE on pathology and missed direct evidence of the role of 4-HNE in respective relevant diseases.
Specific comments
- To provide some example of 4-HNE molecular mechanisms on cancers in section “HNE in pathology”.
- If the authors can elaborate the direct evidence of correlation between 4-HNE level and the trigging of cell death pathway in cancers, this review article will be more more attractive.

Author Response
Reply to Reviewer 2
- To provide some example of 4-HNE molecular mechanisms on cancers in section “HNE in pathology”.
Reviewer 3 suggested that "the subtitle of Section 4 should reflect the focus on HNE as a potential target for apoptosis and cancer therapy". Therefore, the title „A desirable presence of HNE” was changed to „A dual role of HNE in cancer”, where provide some example of 4-HNE molecular mechanisms on cancers.
If the authors can elaborate the direct evidence of correlation between 4-HNE level and the trigging of cell death pathway in cancers, this review article will be more more attractive.
We agree with the Reviewer's suggestion. An appropriate informations - we hope - have been shown.
Taking into account comments of the Reviewer suggesting that English needs improvement we want to inform you that that our paper was checked for proper English usage by a professional Polish-English translator in the field of scientific papers.

Reviewer 3 Report
This mini review article focuses on HNE, a primary aldehyde product from lipid peroxidation and is implicated in various pathologies and biochemical processes. The authors described the chemical process of HNE production and its adduction with macromolecules in relative detail, and briefly summarized its implication in pathologies and potential role in cancer pathology and treatment. Overall, this short article is easy to follow and presents some interesting points in HNE.
Figure1, -OH at carbon 5?
Section 3, a table showing HNE-implicated pathologies/diseases will make it easier to understand.
The subtitle of Section 4 should reflect the focus on HNE as a potential target for apoptosis and cancer therapy.
Line213-216, it is unnecessary to list the authors’ names.
Line 252-253, “10, 25, 200 mM”? check the unit
Author Response
Reply to Reviewer 3
We are grateful to the Reviewer for valuable comments and thorough review of our paper.
Please, find my replies below.
Figure1, -OH at carbon 5?
Thank you to the Reviewer for adequate remark. It has been corrected
Section 3, a table showing HNE-implicated pathologies/diseases will make it easier to understand
We agree with the Reviewer's suggestion. An appropriate table has been made.
The subtitle of Section 4 should reflect the focus on HNE as a potential target for apoptosis and cancer therapy.
We agree with the Reviewer's suggestion. It has been corrected
Line213-216, it is unnecessary to list the authors’ names.
It has been corrected. The authors’ names were removed .
Line 252-253, “10, 25, 200 mM”? check the unit
Thank you to the Reviewer for adequate remark. Of course, they should be micromoles, (μM), not millimoles (mM). It has been corrected.
Taking into account comments of the Reviewer suggesting that English needs improvement we want to inform you that that our paper was checked for proper English usage by a professional Polish-English translator in the field of scientific papers.

Reviewer 4 Report
Comments and suggestions for authors are in the attached file:
biomolecules 1526571

Author Response
Reply to Reviewer 4
We are grateful to the Reviewer for valuable comments and thorough review of our paper.
Please, find my replies below.
- The title is ambiguous. There is a perceived lack of focus. The authors should consider
rethinking the objective of the review. "An attractive molecule", for what? In what sense? What is the focus?
As suggested by the Reviewer, the title of the manuscript has been changed.
- The authors extensively discuss the chemistry of 4-HNE; however, it is important to delve into the biological properties of 4-HNE and its participation in the different pathologies.
- The review would be enriched if the authors expanded the information on the current
knowledge of the mechanisms and signaling pathways through which 4-HNE participates in the different pathologies.
As suggested by the Reviewer, more information has been provided on signal paths through which 4-HNE participates in the different pathologies.
- The phrase "from sinfulness to holiness" is awkward, authors should consider the use of formal or scientific language.
The phrase "from sinfulness to holiness" has been deleted. In this place the phrase was written: “HNE is not only harmful but beneficial, too”.
- The authors must homogenize the use of the terms HNE, 4HNE, or 4-HNE in the
manuscript; this can confuse the readers.
It has been corrected.
- On different occasions, phrases such as: “Thus, in my opinion, the production of lipid
free radicals and HNE is a side effect of LOX and COX action”, “Therefore, I suggest the readers…”, which suggests the opinion of a single author. It is suggested to reach a
consensus considering all the author´s views and conclusions.
It has been corrected.
- The authors should discuss the HNE in pathology section in greater detail and delve into the dual role of 4-HNE in cancer.
It has been done.
- The effects of 4-HNE are a function of concentration. What is the concentration of 4-HNE under physiological conditions, during oxidative stress, and under pathological conditions?
We agree with the Reviewer's suggestion. Relevant data has been provided.
Taking into account comments of the Reviewer suggesting that English needs improvement we want to inform you that that our paper was checked for proper English usage by a professional Polish-English translator in the field of scientific papers.

Round 2
Reviewer 1 Report
The review was strengthened considerably by your corrections. However, I cannot fully agree to the statement that HNE has (been) converted from sinner to saint. Even in your review you point out that HNE has not only modulating, but at least co-carcinogenic properties. Therefore I ask you to reconsider the term "holiness" for HNE.
Author Response
Reply to Reviewer 1
We are grateful to the Reviewer for valuable comments and thorough review of our paper.
Please, find my replies below.
The review was strengthened considerably by your corrections. However, I cannot fully agree to the statement that HNE has (been) converted from sinner to saint. Even in your review you point out that HNE has not only modulating, but at least co-carcinogenic properties. Therefore I ask you to reconsider the term "holiness" for HNE.
It has been corrected.

Reviewer 3 Report
The revised manuscript is improved substantially and satisfied most concerns. It should be accepted after responding to the following concerns.
Line 335, “…lead to various human pathologies” is a very strong statement. The reality is that in most cases, HNE modification of proteins is only found being associated with pathology and we lack the direct evidence of whether it leads to the pathology.
Line 425, “…about 10 times higher than in human plasma”, add citation
Line 513-514, use mild or higher level, instead of increased or decreased
Author Response
Reply to Reviewer 3
We are grateful to the Reviewer for valuable comments and thorough review of our paper.
Please, find my replies below.
The revised manuscript is improved substantially and satisfied most concerns. It should be accepted after responding to the following concerns.
Line 335, “…lead to various human pathologies” is a very strong statement. The reality is that in most cases, HNE modification of proteins is only found being associated with pathology and we lack the direct evidence of whether it leads to the pathology.
I share the Reviewer's opinion. This sentence has been redrafted as suggested by the Reviewer.
Line 425, “…about 10 times higher than in human plasma”, add citation
The relevant literature item has been cited.
Line 513-514, use mild or higher level, instead of increased or decreased
The sentence was corrected as suggested by the Reviewer.

Reviewer 4 Report
The comments and suggestions are in the attached file

Author Response
Reply to Reviewer 4
We are grateful to the Reviewer for valuable comments and thorough review of our paper.
Please, find my replies below.
It is important that the title reflects the content of the review Most of the review addresses chemical aspects, including the lipid peroxidation process, HNE formation. The pathological aspects are little addressed. The therapeutic aspects that are addressed only focus on cancer: Is there evidence in other diseases? Is the effect of HNE as a therapeutic strategy only through the induction of apoptosis? Have other mechanisms been described?
In line with the Reviewer's opinion, I changed the title of our review. The current title is: “Chemistry and biochemistry aspects of the 4-hydroxy-2,3-trans-nonenal”.
Authors must take special care in citations; they must check that the information provided matches the author mentioned and that the reference is appropriate. “Studies of Jaganjac et al. conducted in war veterans with posttraumatic stress disorder (PTSD) revealed significantly elevated levels of plasma HNE in these patients. Moreover, the authors have shown that accumulation of plasma HNE seems to increase with aging but is negatively correlated with BMI, showing a specific pattern of changes for individuals diagnosed with PTSD [43].” The information in the previous paragraph coincides with the following article, which does not appear in the list of references. Perković MN, Milković L, Uzun S, Mimica N, Pivac N, Waeg G, Žarković N. Association of Lipid Peroxidation Product 4-Hydroxynonenal with Post-Traumatic Stress Disorder. Biomolecules. 2021 Sep 15;11(9):1365. doi: 10.3390/biom11091365. PMID: 34572578; PMCID: PMC8469760.
I am sorry for this mistake. Thank you for this remark. It has been corrected.
On line 9: 4-Hydroxy-2,3-trans-nonenal (C9H16O2), also known as 4-hydroxy-2,3-trans nonenal what is the difference?
Thank you for this remark. It has been corrected.
